# Spherical Diffusion Flames of Ethylene in Microgravity: Multidimensional Effects

Sergey M. Frolov [1,2,*], Vladislav S. Ivanov [1,2], Fedor S. Frolov [1,2] and Ilya V. Semenov [2]

1    Department of Combustion and Explosion, Semenov Federal Research Center for Chemical Physics of the Russian Academy of Sciences, Moscow 119991, Russia; vsivanov@chph.ras.ru (V.S.I.); f.frolov@chph.ru (F.S.F.)
2    Department of Computational Mathematics, Federal State Institution "Scientific Research Institute for System Analysis of the Russian Academy of Sciences", Moscow 117218, Russia; ilyasemv@yandex.ru
*    Correspondence: smfrol@chph.ras.ru

**Abstract:** The joint American–Russian Space Experiment Flame Design (Adamant) was implemented on the International Space Station (ISS) in the period from 2019 to 2022. The objectives of the experiment were to study the radiative extinction of spherical diffusion flames (SDF) around a porous burner (PB) under microgravity conditions, as well as the mechanisms of control of soot formation in the SDF. The objects of the study were the normal and inverse SDFs of gaseous ethylene in an oxygen atmosphere with nitrogen dilution at room temperature and pressures ranging from 0.5 to 2 atm. The paper presents the results of transient 1D and 2D calculations of 24 normal and 13 inverse SDFs with and without radiative extinction. The 1D calculations revealed some generalities in the evolution of SDFs with different values of the stoichiometric mixture fraction. The unambiguous dependences of the ratio of flame radius to fluid mass flow rate through the PB on the stoichiometric mixture fraction were shown to exist for normal and inverse SDFs. These dependences allowed important conclusions to be made on the comparative flame growth rates, flame lifetime, and flame radius at extinction for normal and inverse SDFs. The 2D calculations were performed for a better understanding of the various observed non-1D effects like flame asymmetry with respect to the center of the PB, flame quenching near the gas supply tube, asymmetrical flame luminosity, etc. The local mass flow rate of fluid through the PB was shown to be nonuniform with the maximum flow rate attained in the PB hemisphere with the attached fluid supply tube, which could be a reason for the flame asymmetry observed in the space experiment. The evolution of 2D ethylene SDFs at zero gravity was shown to be oscillatory with slow alterations in flame shape and temperature caused by the incepience of torroidal vortices in the surrounding gas. Introduction of the directional microgravity, on the level of $0.01g$, led to the complete suppression of flame oscillations.

**Keywords:** space experiment; microgravity; spherical diffusion flame; ethylene; numerical simulation; radiative flame extinction; multidimensional effects

## 1. Introduction

The joint American-Russian Space Experiment Flame Design (Adamant) was conducted on the International Space Station (ISS) in the period from 2019 to 2022. The present authors were the members of the research team of the Russian Space Agency (Roskosmos). The objective of the space experiment was to study the radiative extinction of the spherical diffusion flame (SDF) around a spherical porous burner (PB) under microgravity conditions. Flame extinction is a critical phenomenon determined by the balance between the rates of chemical heat release in the flame and the radiative heat loss from the flame. This phenomenon is observed solely in microgravity conditions, whereas in terrestrial conditions, it is hindered by buoyancy. The understanding of the flame extinction mechanism is, thus, extremely important for fire safety on spacecraft. In the space experiment, both normal and inverse SDFs of non-premixed gaseous ethylene and oxygen at room temperature and pressure ranging from 0.5 to 2 atm were the objects of the study. The normal and inverse

SDFs were formed when the fluid supplied through the PB was either fuel or oxygen, respectively, whereas the gas in the surrounding atmosphere was either oxygen or fuel, respectively. Both fuel and oxygen could be diluted with nitrogen. Contrary to terrestrial experiments in the 2.2-s drop tower [1], the observation time of the SDFs on the ISS could be two orders of magnitude larger, i.e., could reach 200 s [2,3].

Figure 1 shows examples of images of a sooty and sootless normal (Figures 1a and 1b, respectively) and inverse SDFs (Figures 1c and 1d, respectively) in microgravity conditions [4,5]. Several specific features are worth mentioning. First, such flames are nonstationary. Their size increases with time while the flame temperature decreases, leading to radiative extinction. The phenomenon of radiative extinction was first considered in [6–8], and further studied experimentally, theoretically, and computationally elsewhere [9–16]. Most of the experimental studies were carried out in drop towers with a short observation time, insufficient for the flame to develop independently of the ignition stage. Most of the theoretical and computational studies used simplified descriptions of transport and chemistry with some characteristic Damkoehler, Markstein, and Lewis numbers that could yield qualitatively loose and quantitatively inaccurate results [10]. Extinction occurs at a constant flame temperature ranging from 1100 to 1200 K for hydrogen, methane, ethylene, and propane SDFs [1,10,14,16] when the flame grows to a certain (large) size, and can be sudden or gradual with flame shape and temperature fluctuations [10]. Various diluents affect flame evolution mainly when added on the fuel side [11,13].

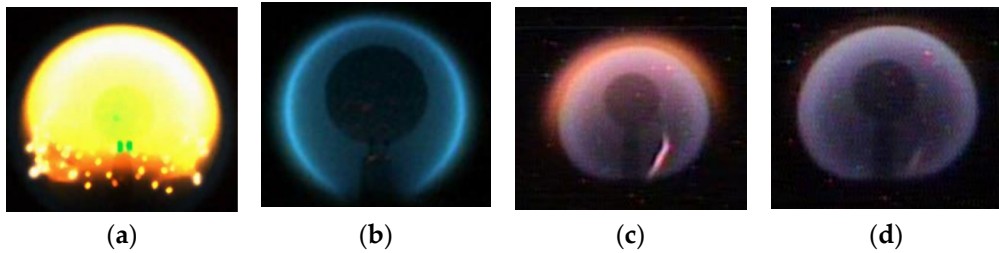

| (**a**) | (**b**) | (**c**) | (**d**) |

**Figure 1.** Photographs of the spherical diffusion flames of ethylene in the flame design (Adamant) space experiment under microgravity conditions on the ISS: (**a**) sooty normal flame; (**b**) sootless normal flame; (**c**) sooty inverse flame; and (**d**) sootless inverse flame [4,5].

Secondly, the SDFs appear to be asymmetrical with respect to the center of the PB (see Figure 1b–d). For some reason, the SDF center is shifted toward the fluid supply tube (from now on, to the "south" pole). Third, SDF luminosity in the southern hemisphere is, for some reason, lower than in the northern hemisphere, so that the flame looks somewhat truncated due to flame quenching near the gas supply tube (see Figure 1a,b). Fourth, as soot is prone to appear on the fuel side [17,18], it forms inside the normal SDF and outside the inverse SDF (see Figure 1a,c) [19–21]. According to [21], soot forms in the region where the local C/O atomic ratio and local temperature satisfy the conditions $0.32 < C/O < 0.44$ and $T > 1300$–$1500$ K.

The phenomenon of radiative flame extinction was also studied for droplet flames [22–26]. Contrary to the SDFs with controlled steady-state fluid supply to the PB, the fuel supply in droplet flames is determined by the rate of liquid vaporization, which is a function of time as well as the properties of liquid and surrounding gas. Interestingly, the extinction of the droplet flame can be followed by secondary flame flashes caused by low-temperature chemical reactions producing active intermediate products like alkyl and hydrogen peroxides. Upon reaching a critical concentration, these intermediate products thermally decompose with the formation of highly active hydroxyl radicals, causing a secondary flash of flame around the droplet. According to [25], the secondary flashes of n-dodecane droplet flames occur at a characteristic flame temperature of 1100 K, while the local instantaneous gas temperature in the region of flash origin is about 980–1000 K.

The objective of this study is to find some generalities in the evolution of normal and inverse SDFs in microgravity conditions using transient one-dimensional (1D) and

two-dimensional (2D) numerical simulations of multiple flames investigated on the ISS. The 2D simulations are focused on revealing the possible reasons for the flame shape and luminosity asymmetry observed in the space experiment.

## 2. Materials and Methods

### 2.1. Statement of the Problem

The statement of the problem is the same as in [21]. Here, we briefly reproduce it.

The PB with radius $R_s$ = 3.2 mm is mounted on a gas supply tube 160 mm long with an inner diameter of 1.5 mm. Figure 2 shows the photograph of the PB (Figure 2a) and the schematics of 2D and 1D computational domains (Figure 2b,c). The volume of the combustion chamber is approximately equal to 90 L (the outer wall radius $r_\infty$ = 288 mm). Initially, at $t = 0$, the chamber is filled with a nitrogen-diluted oxidizer or combustible gas (hereinafter, external gas) with temperature $T_0$, pressure $P_0$, and the $i$th species mass fractions $Y_{i0}$. At $t > 0$, a combustible gas or oxidizer (hereinafter referred to as the supplied gas) with constant mass fractions of the species $Y_{i,\text{in}}$ is delivered through the supply tube to the PB with a constant mass flow rate $G_{\text{in}}$ (hereinafter, index "in" refers to the inlet section of the supply tube). The supplied gas, passing through PB pores, displaces the initial external gas from it and enters the combustion chamber. All studies were performed with ethylene. The kinetics of ethylene oxidation and combustion are well known. The gas supply through the PB is accompanied by molecular mixing of the supplied and external gases and the formation of a combustible mixture in the vicinity of the PB, which is ignited by an external ignition source at time $t_{ign}$. After a certain transition period, an SDF is formed around the PB. In the experiment, the flame radius is determined by the average size of the luminous zone. In 1D calculations, the flame size is defined as the radial coordinate of the maximum gas temperature $R_f$, with the origin of the coordinate in the PB center [21]. During combustion, the position of the flame may change: the flame radius may both increase and decrease. Generally speaking, depending on the governing parameters of the problem ($G_{\text{in}}$, $Y_{i,\text{in}}$, $Y_{i0}$, etc.), situations are possible where $R_f > R_s$ and $R_f \leq R_s$. Combustion in a flame is accompanied not only by chemical energy release, molecular and turbulent transport of mass, momentum, and energy but also by the processes of radiative heat removal to the surrounding space. Radiation is emitted mainly by triatomic molecules ($H_2O$ and $CO_2$) and diatomic molecules ($N_2$ and $O_2$), as well as soot, which can form during combustion.

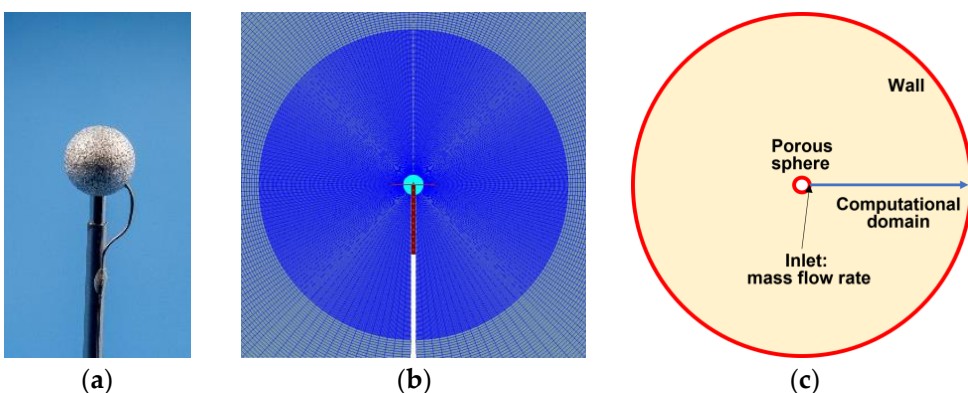

(**a**)             (**b**)             (**c**)

**Figure 2.** Photograph of the porous burner mounted on a gas supply tube (**a**) and the schematics of the 2D (**b**) and 1D (**c**) computational domain.

The physical and mathematical model must include all the most important processes involved in the described phenomenology. Since the Navier–Stokes equations are valid for both nonporous and porous media, for solving the problem, we apply the general

three-dimensional (3D) time-dependent Reynolds-averaged Navier–Stokes, energy, and species conservation equations for a multicomponent reacting gas:

$$\rho\frac{\partial U_i}{\partial t} + \rho U_j\frac{\partial U_i}{\partial x_j} = -\frac{\partial p}{\partial x_i} + \frac{\partial}{\partial x_j}\left[\tau_{ij} + \tau_{ij}^t\right] + \rho g_i;\tag{1}$$

$$\rho\frac{\partial I}{\partial t} + \rho U_j\frac{\partial I}{\partial x_j} = \rho\dot{Q} + \frac{\partial p}{\partial t} + \frac{\partial}{\partial x_j}\left[\left(\tau_{ij} + \tau_{ij}^t\right)U_j\right] + \frac{\partial}{\partial x_j}\left(q_j + q_j^t\right) + \Omega;\tag{2}$$

$$\rho\frac{\partial Y_l}{\partial t} + \rho U_j\frac{\partial Y_l}{\partial x_j} = \rho\dot{w}_l + \frac{\partial}{\partial x_j}\left[\rho\left(j_{lj} + j_{lj}^t\right)\right],\tag{3}$$

where, from now on, standard notations are used (see nomenclature). The turbulent fluxes of momentum, energy, and species in Equations (1)–(3) can be modeled, e.g., within the framework of a certain turbulence model. The governing equations are closed by the caloric and thermal equations of state, the reaction mechanism of fuel oxidation and soot formation, the relationships for the fluxes and source terms, as well as by the initial and boundary conditions.

According to the phenomenology, the following simplifying assumptions can be adopted:

1.  In microgravity conditions, the acceleration of gravity $g_i$ is negligible;
2.  The evolution of the SDF is symmetrical with respect to the axis of the supply tube;
3.  The porous medium in the flow region can be modeled by flow resistance according to the Darcy law and heat exchange with the fluid according to the Newton law, i.e., the porous medium can be represented by added momentum and heat sources, $\left(\frac{\partial P}{\partial x_i}\right)_s$ and $\Psi_s$ respectively, in the governing equations. In addition, since the porous medium reduces the volume accessible for fluid, the local flow velocity, $U_i$ and superficial velocity inside the porous medium, $u_i$ are coupled by the undirected porosity value $\varphi$: $U_i = \varphi u_i$;
4.  The thermophysical and structural parameters of the PB material are constant;
5.  Thermal radiation of PB is negligible; PB absorbs thermal radiation of soot, $H_2O$, $CO_2$, and (for the sake of generality) $N_2$ and $O_2$;
6.  Gas-phase and catalytic reactions in the PB are absent;
7.  The gas flow is laminar;
8.  A multicomponent gas mixture obeys the thermal and caloric equations of state of an ideal gas and possesses variable thermophysical properties;
9.  The effect of thermodiffusion is negligible;
10. Soot is an equivalent gas with the molecular mass of atomic carbon, when simulating soot reactions;
11. Soot particles are the clusters of 20–25 carbon atoms, have the corresponding constant size, and do not coagulate;
12. The radiation heat flux is caused solely by soot, $H_2O$, $CO_2$, $N_2$, and $O_2$ emittance;
13. The outer wall of the computational domain is impermeable, noncatalytic, and isothermal.

Following assumption 2, the computational domain can be represented in the form of a 2D sector with the axis along the supply tube and a small opening angle divided into control volumes in the axial and radial directions (see Figure 2b). Following assumptions 3 to 6, the energy conservation equation for the porous medium reads:

$$\rho_s c_s\frac{\partial T_s}{\partial t} = \lambda_s\frac{\partial}{\partial x_j}\left(\frac{\partial T_s}{\partial x_j}\right) + \Omega_s;\tag{4}$$

where $T_s$ is the temperature of the PB and $\Omega_s$ is the heat source/sink for the PB. Moreover, assumptions 3 and 5 mean that the term $\Omega_s$ in Equation (4) contains only two contributions: $\Psi_s$ and the radiation absorption $\Omega_{sg}$, whereas assumption 4 implies that the PB radius $R_s$ is constant. Following assumption 7, the turbulent fluxes in Equations (1)–(3) can be

omitted, i.e., $\tau_{ij}^t = q_j^t = j_{lj}^t = 0$. Assumption 8 is conventional and implies the validity of the equations of state:

$$p = \rho RT \sum_{l=1}^{N} \frac{Y_l}{W_l} \tag{5}$$

and

$$H_l = H_l^0 + \int_{T^0}^{T} c_{p,l} dT \tag{6}$$

Assumption 9 means that molecular fluxes $q_j$ and $j_{lj}$ can be determined as:

$$q_j = -\left( \lambda \frac{\partial T}{\partial x_j} \right), \; j_{lj} = -D_l \frac{\partial Y_l}{\partial x_j}, l = 1, \dots, N-1 \tag{7}$$

where $\lambda$ is the thermal conductivity and $D_l$ is the effective diffusion coefficient of the $l$th species in the mixture [27].

A simple macrokinetic mechanism of soot formation and oxidation [28] including four irreversible overall reactions is used. In the mechanism, soot C is considered a gaseous species in the reaction mechanism with its own mass fraction, $Y_{soot}$, according to assumption 10. The mechanism uses acetylene $C_2H_2$ as a precursor of soot C. Except for soot C, all other substances involved in the reactions of soot formation and oxidation are included in the detailed reaction mechanism (DRM) of ethylene oxidation [29], containing 48 species and 209 reversible elementary reactions. The kinetic parameters of reactions, namely, the pre-exponential factor, $A_k$, activation energy $E_k$, and the temperature exponent in the expression for the rate of the $k$th reaction, were determined using the thoroughly tested DRM of soot formation [30] from the condition of the best agreement between the results of calculations for the soot yield obtained on the basis of DRM and on the basis of the macrokinetic mechanism. For estimating the effect of soot radiation, it is conditionally assumed (assumption 11) that soot particles possess the specific (per unit mass) emitting surface, $S_{soot} = 6/(d_{soot}\rho_{soot})$ (here $d_{soot}$ is the conditional soot particle size, $\rho_{soot}$ is the soot density), which is directly connected to the soot mass fraction $Y_{soot}$. Thus, the simplest radiation model is used without radiation reabsorption and scattering by soot particles.

Assumption 12 implies that the source term $\Omega$ in Equation (2) contains the contribution $\Omega_{soot}$ proportional to $S_{soot}$. Assumption 12 also implies that the term $\Omega$ in Equation (2) contains the contribution $\Omega_g$ proportional to the local instantaneous volume fractions of $H_2O$, $CO_2$, $N_2$, and $O_2$. Assumption 13 is conventional.

If one additionally assumes that the gas supply tube does not affect the evolution of the SDF, and all physicochemical processes are spherically symmetric, then the 1D approximation will be valid. In this case, the computational domain can be represented in the form of a central sector with a small opening angle divided into control volumes in the radial direction (see Figure 2c).

Thus, the 3D governing Equations (1)–(4) are greatly simplified and can be solved using an available CFD code for axisymmetric reactive flows. The explicit form of the source terms entering Equations (1)–(4) are as follows:

$$\dot{w}_l = W_l \sum_{k=1}^{L} (v_{l,k}'' - v_{l,k}') A_k T^{n_k} exp\left( -\frac{E_k}{RT} \right) \prod_{j=1}^{N} \left( \frac{Y_j \rho}{W_j} \right)^{v_{j,k}'} \tag{8}$$

$$\dot{Q} = \sum_{l=1}^{N} H_l \dot{w}_l \tag{9}$$

$$\Omega = \Omega_{soot} + \Omega_g + \Psi_s \tag{10}$$

$$\Omega_{soot} = \sigma S_{soot} Y_{soot} \rho \left( T^4 - T_0^4 \right) \tag{11}$$

$$\Omega_g = \sigma p \sum_{l=1}^{4} a_l(T) X_l \left( T^4 - T_0^4 \right) \tag{12}$$

$$\Psi_s = \alpha_s S_{\mathrm{PB}}(T - T_s) \tag{13}$$

$$\Omega_s = \Psi_s + \Omega_{sg} \tag{14}$$

$$\Omega_{sg} = \delta_s \varepsilon_s \int \left( \Omega_{soot} + \Omega_g \right) dV \tag{15}$$

$$\left( \frac{\partial p}{\partial x_i} \right)_s = \left( \frac{\partial p}{\partial x} \right)_s = \frac{\mu}{\kappa} u \tag{16}$$

where $S_{\mathrm{PB}} = 6(1 - \varphi)/d$ is the specific surface area of the PB and $\delta_s$ is the delta function used to account for radiation absorption only on the PB surface. Equation (16) is used according to recommendations [31]. The values of the coefficients $a_l$ for $H_2O$ and $CO_2$ are taken from the polynomials in [32]; for $N_2$ and $O_2$, $a_l$ is independent of the gas temperature and is assumed to be equal to 0.1. The transport coefficients of the gas, as well as the effective diffusion coefficients of species in the gas mixture and specific heats, are calculated by the formulae presented in [33].

Figure 3 shows some important dimensions of the computational domain, as well as the initial and boundary conditions. The computational domain is represented by a cylindrical sector with a solid angle of 3°. The supply tube is treated as isothermal (293 K) or adiabatic. In the beginning, the supply tube and a void in the PB are filled with either fuel and nitrogen (normal flame) or oxygen and nitrogen (inverse flame) at normal pressure and temperature (NPT) conditions ($p = p_0, T = T_0$) and the rest of the computational domain is filled with oxygen and nitrogen (normal flame) or fuel and nitrogen (inverse flame) at NPT conditions. The boundary condition at the inlet of the supply tube is the preset mass flow rate of fluid, its temperature (293 K), and composition:

$$\rho U S_{\mathrm{in}} = G_{\mathrm{in}}, T = T_0, Y_l = Y_{l0}(l = C_2H_4, N_2), Y_l = 0(l \neq C_2H_4, N_2)$$

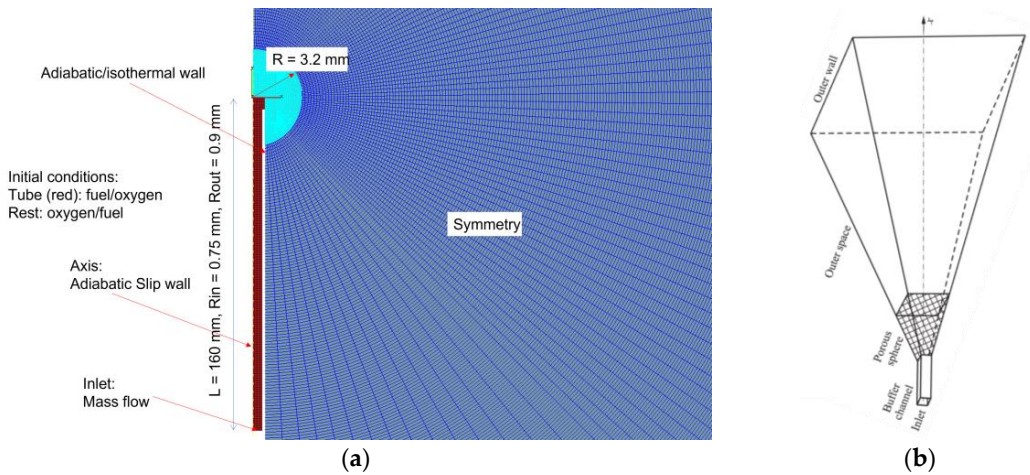

**Figure 3.** Some dimensions of the 2D (**a**) and 1D (**b**) computational domains, as well as initial and boundary conditions.

The boundary conditions at all rigid walls, including the outer boundary of the computational domain, are:

$$U = 0, T = T_0, \frac{\partial Y_l}{\partial x} = 0, l = 1, \ldots, N$$

Symmetry conditions are satisfied on the side surfaces of the computational domain.

The task of the calculations is to find some generalities in the evolution of normal and inverse SDFs in microgravity conditions using transient 1D and 2D numerical simulations of multiple flames investigated on the ISS, as well as to demonstrate the multidimensional effects on the spatial structure of the SDF and its evolution in time. It is worth noting that, in accordance with the theory, an important parameter of diffusion flames is the stoichiometric mixture fraction $0 < Z_{st} < 1$ determined by the mass fractions of the oxidizer $Y_O$ and the fuel $Y_F$ in the burner:

$$Z_{st} = \frac{Y_O}{Y_O + \eta Y_F}$$

where $\eta = \nu_O W_O / \nu_F W_F$, $\nu_O = 3$, and $\nu_F = 1$ are the stoichiometric coefficients in the overall reaction equation $C_2H_4 + 3O_2 = 2CO_2 + 2H_2O$, whereas $W_O = 32$ kg/kmol and $W_F = 28$ kg/kmol are the molecular masses of oxygen and ethylene. It is generally believed that flames with low $Z_{st}$ are more prone to soot formation than flames with large $Z_{st}$ [17,18].

*2.2. Numerical Solution*

The 2D computational domain of Figure 3a contains about 500,000 cells with the characteristic size of less than 0.1 mm near the porous sphere. Dark red, cyan, and blue colors in Figure 3a correspond to the cells of the supply tube, porous sphere, and surrounding gas volume, respectively. Grid-sensitivity studies showed that the adopted spatial resolution was sufficient for obtaining grid-independent results.

The 1D computational domain of Figure 3b contains a planar buffer channel with a cross-sectional area $S_{in}$, PB, and an outer space bounded by a wall. The planar buffer channel of length $R_0 = 10$ mm is used to provide undisturbed boundary conditions. The buffer channel is immersed in the PB so that the area of the inlet section in the PB is equal to $S_{in}$. Symmetry conditions are satisfied on the side surfaces of the computational domain. The entire computational domain is divided into 1333 control volumes compressed towards the outer surface of the PB, which ensures the grid independence of the obtained results.

The set of governing Equations (1)–(4) with additional relations (5) to (16) is solved by the control volume method using the segregated SIMPLE/PISO algorithm (see [3,21]). Convective transfer in the law of conservation of mass is approximated by the central difference and in the law of conservation of momentum by the total variation diminishing (TVD) scheme with the MINMOD limiter. For the finite-volume approximation of all other equations, the standard UPWIND scheme of the first order is used.

Before calculating the dynamics of SDF, a proper ignition procedure was developed. The main requirement of the ignition procedure was the weakest possible effect on the flow field caused by a localized pressure rise. Preliminary calculations showed that the stronger the ignition, the later the arising flow field became insensitive to ignition. The resulting procedure in 1D calculations is as follows. In a time interval of 0.2–0.3 s after the start of the gas supply to the PB, in the region $x \in [r_s + 0.0005 \text{ m}, r_s + 0.0060 \text{ m}]$, the mixture temperature is instantaneously changed to $T_{ign} = 1300$ K. The time interval and region are selected to allow for the formation of a small portion of flammable premixed fuel–oxidizer composition. The ignition temperature of 1300 K is found to be about the minimal temperature required for mixture ignition. In the 2D calculations, ignition was simulated by imposing the 1D fields of velocity, temperature, pressure, and species concentrations on the 2D mesh. These fields were taken from the 1D calculation at a time instant when the flame was seemingly established (about 10 s, [21]).

The following values of the governing parameters are used in the calculations: $T_0 = 293$ K, $p_0 = 0.1$ MPa (unless otherwise specified), $S_{in} = 5 \cdot 10^{-10}$ m$^2$, $R_s = 0.0032$ m, $r_\infty = 0.288$ m, $\varphi = 0.5$, $\kappa = 10^{-13}$ m$^2$; $d = 10^{-5}$ m; $\varepsilon_s = 0.8$, $\rho_s = 4000$ kg/m$^3$, $c_s = 650$ J/(kg·K), $\lambda_s = 5$ W/(m·K), $d_{soot} = 2$ nm, $\rho_{soot} = 2000$ kg/m$^3$, $N = 49$, $L = 213$.

## 3. Results and Discussion

### 3.1. Experiments

The experimental apparatus and measurement techniques are described in detail in [2]. Tables 1 and 2 show the conditions for some selected experiments with normal and inverse flames, respectively. The gray fill marks the flames for which radiative extinction was observed in experiments and/or in calculations. The experiments were carried out both with pure ethylene and with ethylene diluted with nitrogen. Oxygen was always diluted with nitrogen for safety reasons. Ethylene and nitrogen were supplied either to the PB (normal flames) or to the combustion chamber (inverse flames) from different cylinders without premixing. The instantaneous flow rates of both gases were controlled by flow meters. This approach made it possible to arbitrarily control the degree of ethylene dilution. During the space experiment, no samples of soot or gas were taken. All information on the experiment was sent to the Mission Control Center via communication channels. The mass flow rate of fluid through the PB varied from 0.66 to 8.78 mg/s for normal flames and from 2.3 to 10.1 mg/s for inverse flames, i.e., in nearly similar intervals. The value of $Z_{st}$ varied from 0.06 to 0.41 for normal flames and from 0.20 to 0.76 for inverse flames, i.e., in overlapping intervals. The columns in Tables 1 and 2 show the flame number, flame identification number, volume fractions of gases in the combustion chamber ($X_{O2}$, $X_{N2}$) and in the PB ($X_{C2H4}$, $X_{N2}$), mass flow rate of fluid through the PB ($G_{in}$), pressure ($p$), and stoichiometric mixture fraction ($Z_{st}$). The last three columns show the values of the measured ($R_{fe,exp}$) and calculated ($R_{fe,calc}$) flame radii and calculated flame temperature ($T_{fe,calc}$) at radiative extinction. As seen from the tables, the 1D model of Section 2.1 predicts the flame radii at extinction reasonably well for both normal and inverse flames. In general, the predicted critical flame temperature at extinction (about 1200 K) also corresponds well with measurements. As an example, Figures 4 and 5 compare calculations with measurements for normal flame No. 4 (19189K1, Figure 4) and inverse flame No. 29 (22018G3, Figure 5). In the figures, flame extinction occurs at the time instant corresponding to the last experimental point and to the kink at the curve. The ordinates of the last experimental point and the kink at the curve in Figures 4a and 5a correspond reasonably well with each other.

**Table 1.** Conditions for some normal flames.

| No. | Flame | Combustion Chamber | | Porous Burner | | | $p$, atm | $Z_{st}$ | $R_{fe,exp}$, mm | $R_{fe,calc}$, mm | $T_{fe,calc}$, K |
|---|---|---|---|---|---|---|---|---|---|---|---|
| | | $X_{O2}$ | $X_{N2}$ | $X_{C2H4}$ | $X_{N2}$ | $G_{in}$, mg/s | | | | | |
| 1 | 19115B1 | 0.203 | 0.797 | 1.000 | 0.000 | 0.660 | 1.020 | 0.062 | - | - | - |
| 2 | 19206L6 | 0.203 | 0.797 | 0.291 | 0.709 | 1.980 | 1.010 | 0.184 | - | - | - |
| 3 | 19171D4 | 0.376 | 0.624 | 1.000 | 0.000 | 2.520 | 1.010 | 0.106 | 32 | 28.0 | 1200 |
| 4 | 19189K1 | 0.374 | 0.626 | 0.502 | 0.498 | 5.010 | 1.310 | 0.191 | 31 | 27.5 | 1200 |
| 5 | F10 | 0.380 | 0.620 | 1.000 | 0.000 | 1.224 | 1.239 | 0.107 | - | - | - |
| 6 | F02 | 0.391 | 0.609 | 0.288 | 0.712 | 1.800 | 1.190 | 0.300 | - | - | - |
| 7 | F08 | 0.386 | 0.614 | 0.288 | 0.712 | 3.603 | 1.263 | 0.297 | - | - | - |
| 8 | F05 | 0.400 | 0.600 | 0.288 | 0.712 | 4.514 | 1.250 | 0.304 | - | - | - |
| 9 | 19156C2 | 0.366 | 0.634 | 1.000 | 0.000 | 2.529 | 1.040 | 0.104 | 32 | 28.0 | 1200 |
| 10 | 19142J3 | 0.356 | 0.644 | 1.000 | 0.000 | 2.529 | 0.990 | 0.101 | 31 | 28.0 | 1200 |
| 11 | 19150N1 | 0.296 | 0.704 | 0.168 | 0.832 | 4.885 | 1.010 | 0.360 | - | - | - |
| 12 | 19150G3 | 0.338 | 0.662 | 0.288 | 0.712 | 8.779 | 1.050 | 0.272 | 35.5 | 30.5 | 1200 |
| 13 | 19175A3 | 0.391 | 0.609 | 1.000 | 0.000 | 1.960 | 1.270 | 0.110 | - | - | - |
| 14 | 19206A5 | 0.207 | 0.793 | 0.288 | 0.712 | 8.779 | 1.010 | 0.189 | 31 | 34 | 1200 |
| 15 | 19206G1 | 0.205 | 0.795 | 1.000 | 0.000 | 2.529 | 1.010 | 0.062 | 32 | 32 | 1200 |
| 16 | 19206G4 | 0.201 | 0.799 | 1.000 | 0.000 | 0.822 | 1.010 | 0.061 | - | - | - |
| 17 | 19206L4 | 0.195 | 0.805 | 0.288 | 0.712 | 2.835 | 1.010 | 0.180 | 19 | 19 | 1200 |
| 18 | 19123F1 | 0.206 | 0.794 | 0.490 | 0.510 | 2.640 | 1.010 | 0.120 | 23 | 24 | 1200 |
| 19 | 19123L1 | 0.202 | 0.798 | 1.000 | 0.000 | 2.510 | 1.010 | 0.061 | 33 | 32 | 1200 |
| 20 | 19189J3 | 0.378 | 0.622 | 0.502 | 0.498 | 5.010 | 1.300 | 0.192 | 31 | 28 | 1200 |
| 21 | 19200H3 | 0.285 | 0.715 | 0.131 | 0.869 | 4.430 | 1.020 | 0.411 | - | - | - |
| 22 | 19115F1 | 0.204 | 0.796 | 0.292 | 0.708 | 2.180 | 1.040 | 0.184 | - | - | - |
| 23 | 19123A2 | 0.209 | 0.791 | 1.000 | 0.000 | 1.620 | 1.000 | 0.063 | 25 | 26 | 1200 |
| 24 | 19123C1 | 0.207 | 0.793 | 0.290 | 0.710 | 4.460 | 1.000 | 0.188 | 24 | 25 | 1200 |

**Table 2.** Conditions for some inverse flames.

| No. | Flame | Combustion Chamber | | Porous Sphere | | | $p$, atm | $Z_{st}$ | $R_{fe,exp}$, mm | $R_{fe,calc}$, mm | $T_{fe,calc}$, K |
|---|---|---|---|---|---|---|---|---|---|---|---|
| | | $X_{C2H4}$ | $X_{N2}$ | $X_{O2}$ | $X_{N2}$ | $G_{in}$, mg/s | | | | | |
| 25 | 21328D1 | 0.257 | 0.743 | 0.212 | 0.788 | 10.05 | 1.03 | 0.211 | - | - | - |
| 26 | 21349M3 | 0.270 | 0.730 | 0.212 | 0.788 | 9.11 | 1.00 | 0.203 | - | - | - |
| 27 | 22018H2 | 0.097 | 0.903 | 0.497 | 0.503 | 6.37 | 1.01 | 0.614 | 24 | 20 | 1200 |
| 28 | 22018J1 | 0.096 | 0.904 | 0.318 | 0.682 | 9.73 | 1.01 | 0.514 | 25 | 21 | 1200 |
| 29 | 22018G3 | 0.098 | 0.902 | 0.850 | 0.150 | 7.89 | 1.01 | 0.720 | 31 | 28 | 1200 |
| 30 | 22018G2 | 0.099 | 0.901 | 0.850 | 0.150 | 5.90 | 1.01 | 0.718 | 28 | 24 | 1200 |
| 31 | 22018G1 | 0.099 | 0.901 | 0.850 | 0.150 | 3.90 | 1 | 0.718 | - | - | - |
| 32 | 21328N5 | 0.080 | 0.920 | 0.850 | 0.150 | 2.27 | 0.96 | 0.759 | - | - | - |
| 33 | 22035J2 | 0.096 | 0.904 | 0.850 | 0.150 | 5.90 | 0.51 | 0.725 | - | - | - |
| 34 | 21340M1 | 0.121 | 0.879 | 0.850 | 0.150 | 9.22 | 1.01 | 0.676 | - | 29 | 1200 |
| 35 | 21349N3 | 0.251 | 0.749 | 0.212 | 0.788 | 9.10 | 0.52 | 0.215 | - | - | - |
| 36 | 21349N4 | 0.246 | 0.754 | 0.212 | 0.788 | 10.03 | 0.52 | 0.218 | - | - | - |
| 37 | 22018B1 | 0.168 | 0.832 | 0.850 | 0.150 | 4.7 | 1 | 0.601 | - | - | - |

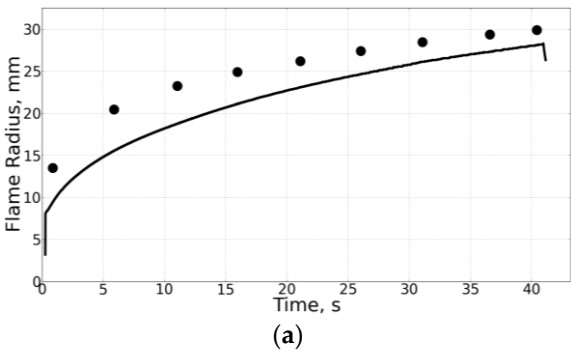 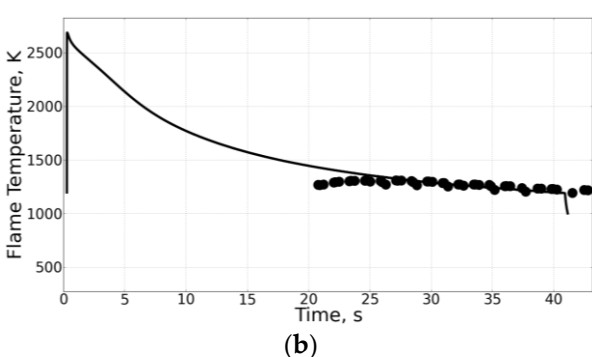

(a)  (b)

**Figure 4.** Comparison of predicted (curve) and measured (black circles) flame radius (**a**) and temperature (**b**) for the normal flame No. 4 (19189K1).

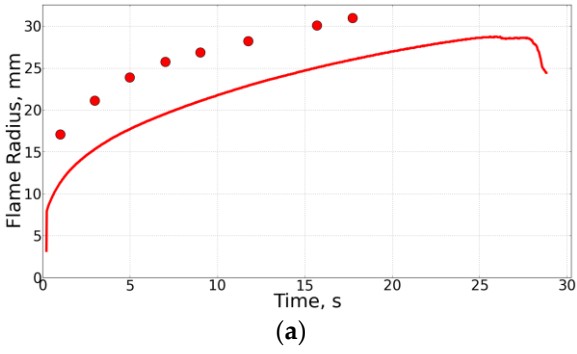 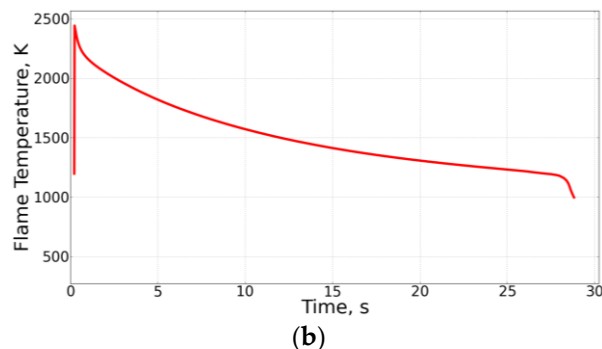

(a)  (b)

**Figure 5.** Comparison of predicted (curve) and measured (red circles) flame radius (**a**) and temperature (**b**) for the inverse flame No. 29 (22018G3). Temperature measurements are not available.

### 3.2. Calculations

#### 3.2.1. 1D Calculations

The objective of 1D calculations was to establish some generalities in the behavior of normal and inverse flames. On the one hand, both experiments and calculations show that the flame radius $R_f$ gradually increases with time for both normal and inverse flames. On the other hand, at the same time instant after ignition, the flame radius is larger for the larger mass flow rate $G_{in}$ of the fluid injected through the porous sphere. In view of it, we have plotted in Figure 6 the dependence of quantity $R_f / G_{in}$ on $Z_{st}$ for all flames listed in Tables 1 and 2 for five instants of time (10 s, 20 s, 30 s, 40 s, and 50 s) after ignition. The

first time instant of 10 s was chosen based on the findings in [21]; in 10 s after ignition, the flame structure does not depend on the ignition process anymore because the soot formed during ignition is either oxidized (normal flames) or moved away from the flame (inverse flames). The plots of Figure 6 show only flames which are not extinguished by the corresponding time instant. Several important observations come from the analysis of Figure 6. First, normal and inverse flames are definitely separated on the $R_f/G_{in}$—$Z_{st}$ plane. Second, there exist the unambiguous dependences of $R_f/G_{in}$ vs. $Z_{st}$ for normal and inverse flames. To demonstrate these dependences, we have fitted all black circles for normal flames in Figure 6a by curve N and all red circles for inverse flames by curve I. The corresponding exponential approximations are shown in Figure 6a. As is seen, all black circles for normal flames are grouped around curve N, whereas all red circles for inverse flames are grouped around curve I. Third, the value of $R_f/G_{in}$ (or flame radius $R_f$) changes with time. To demonstrate the temporal evolution of the $R_f/G_{in}$ values, we have overlayed curves N and I from Figure 6a to Figure 6b–e. As is seen, all black circles in Figure 6b,e for normal flames are still grouped together but move above curve N and all red circles for inverse flames are still grouped together but move above curve I at all time instants from 20 to 50 s, thus indicating that the flame size somehow grows with time. Note that, with time, some flames exhibit radiative extinction and are not present in the figures anymore.

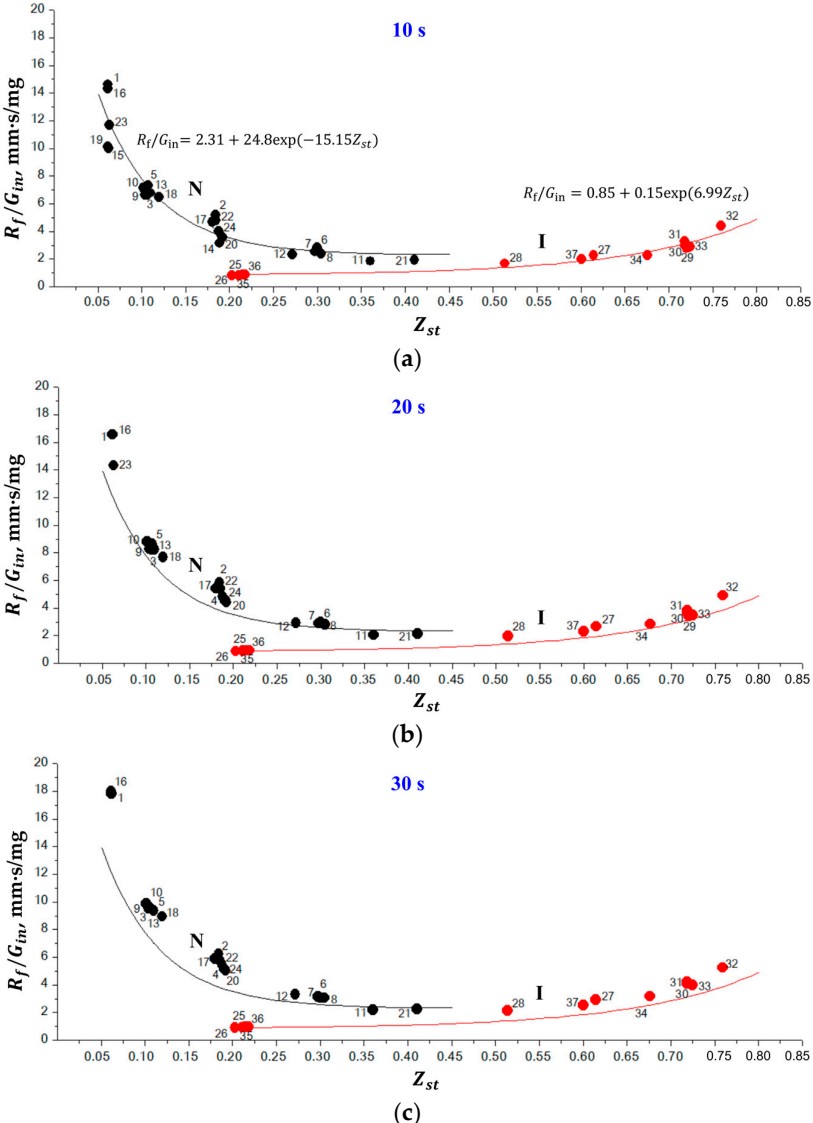

**Figure 6.** *Cont.*

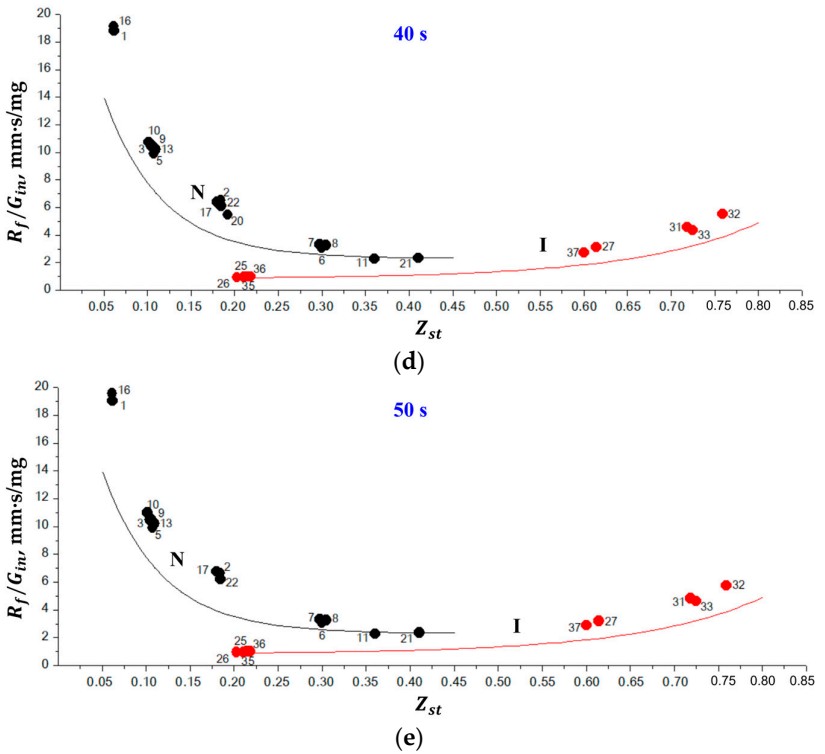

**Figure 6.** Calculated dependences of $R_f/G_{in}$ vs. $Z_{st}$ for normal (black circles) and inverse (red circles) flames at five instants of time: (**a**) 10 s, (**b**) 20 s, (**c**) 30 s, (**d**) 40 s, and (**e**) 50 s after ignition. Numbers correspond to the flame numbers in Tables 1 and 2. Curves N and I correspond to the best fits of calculated data for normal and inverse flames at 10 s after ignition.

The normal flames with the smallest $Z_{st}$ values ($Z_{st} = 0.061$–$0.063$) have the largest values of $R_f/G_{in} = 10$–$15$ mm·s/mg at 10 s and approach 20 mm·s/mg at 50 s. The increase in $Z_{st}$ results in the exponential decrease in the value of $R_f/G_{in}$: at $Z_{st} = 0.411$, $R_f/G_{in} \approx 2$ mm·s/mg for a time ranging from 10 to 50 s. This means that the growth rate of normal flames decreases with $Z_{st}$ and they become more prone to be stationary. This clearly follows from the decreasing deviation of black circles from curve N with the increase in $Z_{st}$ in Figure 6e. At $Z_{st} = 0.36$–$0.411$, the normal flame becomes almost stationary, i.e., its growth rate becomes very low. Contrary to normal flames, for inverse flames, the smallest values of $R_f/G_{in}$ are attained at the smallest $Z_{st}$ values ($Z_{st} = 0.203$, $R_f/G_{in} \approx 1$ mm·s/mg for a time ranging from 10 to 50 s). The increase in $Z_{st}$ results in the exponential increase in the value of $R_f/G_{in}$: at $Z_{st} = 0.759$, $R_f/G_{in} \approx 5$ mm·s/mg at 10 s, and 6 mm·s/mg at 50 s. This means that the growth rate of inverse flames increases with $Z_{st}$ and they become less prone to be stationary. This clearly follows from the increasing deviation of red circles from curve I with the increase in $Z_{st}$ in Figure 6e. At $Z_{st} = 0.203$–$0.218$, the inverse flame becomes almost stationary, i.e., its growth rate becomes very low. Note that the question of the existence of stationary normal and inverse SDFs has not yet been resolved and is currently discussed in the literature [2]. It is also worth noting that the maximum growth rate of inverse flame at $Z_{st} = 0.759$ is considerably less than that of normal flames at $Z_{st} = 0.061$–$0.063$ (see Figure 6e). This means that the lifetime of inverse flames must be longer than that of the normal flames at the identical value of $R_f/G_{in}$.

Interestingly, the extinguished normal and inverse flames with the radius at extinction $R_{fe}$ are also grouped around separate curves in the $R_{fe}/G_{in}$—$Z_{st}$ plane, as shown in Figure 7. The corresponding exponential approximations are shown in Figure 7, together with curves N and I from Figure 6a. Nearly all black and red circles in Figure 7 are seen to lie above curves N and I, respectively, which indicates that the corresponding flames extinguish later than 10 s after ignition.

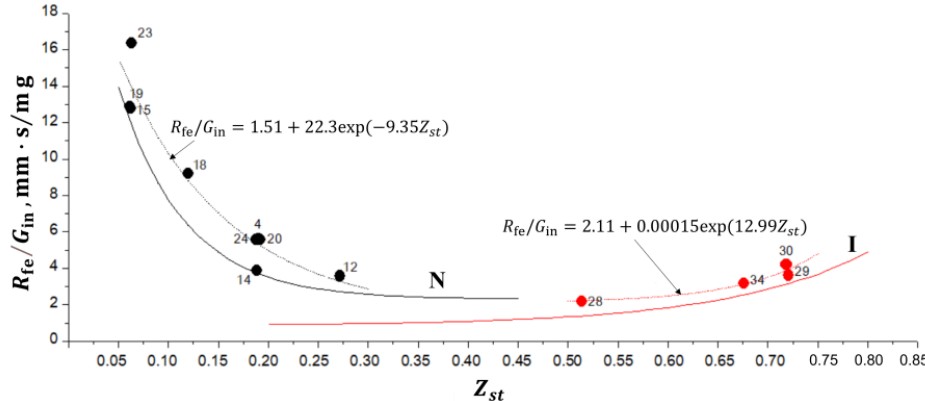

**Figure 7.** Calculated dependences of $R_{fe}/G_{in}$ vs. $Z_{st}$ for normal (black circles) and inverse (red circles) flames at extinction. Numbers correspond to the flame numbers in Tables 1 and 2. Curves N and I correspond to the best fits of the calculated data for normal and inverse flames at 10 s after ignition (see Figure 6a).

The normal flames with the smallest $Z_{st}$ values ($Z_{st}$ = 0.061–0.063) have the largest values of $R_{fe}/G_{in}$ = 13–17 mm·s/mg at extinction. The increase in $Z_{st}$ results in the exponential decrease in the value of $R_{fe}/G_{in}$. The smallest value of $R_{fe}/G_{in} \approx 3.5$ mm·s/mg at extinction is attained at $Z_{st} = 0.272$. For inverse flames, the smallest values of m at extinction are attained at the smallest $Z_{st}$ value ($Z_{st} = 0.514$). The increase in $Z_{st}$ results in the exponential increase in the value of $R_{fe}/G_{in}$: at $Z_{st}$ = 0.718–0.720, $R_{fe}/G_{in} \approx 4$ mm·s/mg. Thus, the maximum values of $R_{fe}/G_{in}$ for normal flames at extinction are a factor of 3–4 larger than for inverse flames. As an example, let us compare the normal flame No. 14 ($Z_{st} = 0.189$) and inverse flame No. 29 ($Z_{st} = 0.720$) with nearly the same values of $R_{fe}/G_{in} \approx 3.5$ mm·s/mg (see Figure 7). According to Tables 1 and 2, these flames have close (large) values of $G_{in}$ (8.779 mg/s vs. 7.89 mg/s). This means that both flames have a close size $R_{fe}$ at extinction. However, due to the lower growth rate, the extinction of the inverse flame occurs later than the extinction of the normal flame, which is confirmed by both 1D calculations (see Figure 6a,b) and the experiment.

The authors of [16] proposed a simple theory of radiative extinction of SDF, in which a fraction of the radiative heat-loss rate from the flame at a critical extinction temperature is equated to the total heat-release rate. Figure 8 compares our results with the theory of [16] at the plane $R_{fe}/R_s$—$G_{in}$. As seen in the wide range of $G_{in}$, the flame radius at extinction is approximately constant and $R_{fe}/R_s \approx 9$, while the theoretical curve underestimates the $R_{fe}/R_s$ values.

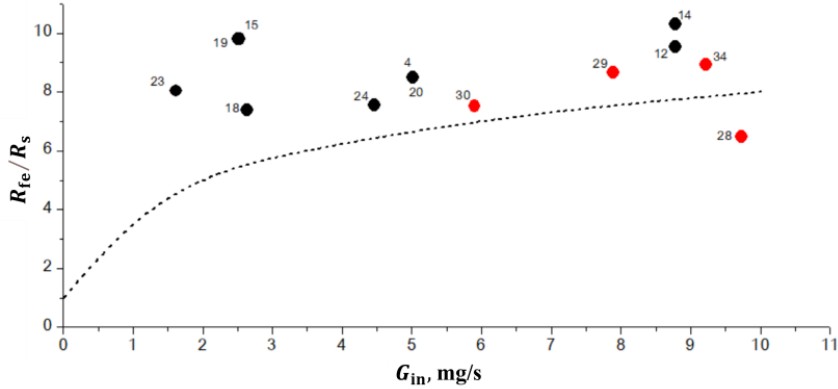

**Figure 8.** Scaled flame radius at extinction as function of inlet mass flow rate for normal (black circles) and inverse (red circles) ethylene flames. Numbers correspond to the flame numbers in Tables 1 and 2. Dashed line corresponds to Equation (4) in [16].

### 3.2.2. 2D Calculations

As mentioned earlier in this paper, for 2D calculations, we use the same code and the same settings as in 1D calculations. Therefore, the results of 2D and 1D calculations for the spherically symmetrical statement of the problem are identical. The only difference between 2D and 1D calculations discussed below in this paper is the extension of the 2D computational mesh to include the gas supply tube, while the mesh around the PB is essentially the same.

Cold Flow

Figure 9 shows the cold flow streamlines from the PB. The streamlines indicate that the flow through the PB is nonuniform and the entire flow pattern is lacking a spherical symmetry. Figure 10a,b show the normalized local mass flow rate (Figure 10a) and normalized accumulated mass flow rate of fluid (Figure 10b) through the PB as functions of the angle in the polar section of the sphere: $-90°$ corresponds to the south pole, $0°$ corresponds to the equator, and $+90°$ corresponds to the north pole of the sphere. The supply tube is seen to block a part of the surface of the PB in the vicinity of the south pole in the angular interval $[-90°, -74°]$. Due to this blockage, the normalized accumulated mass flow rate of fluid through the southern hemisphere is about 15% less than through the northern hemisphere. The local mass flow rate of fluid is nonuniform with the maximum flow rate attained in the angular interval $[-20°, -40°]$ in the southern hemisphere. This nonuniformity in the local mass flow rate of fluid through the PB could be a reason for the flame asymmetry with respect to the center of the PB observed in the space experiment.

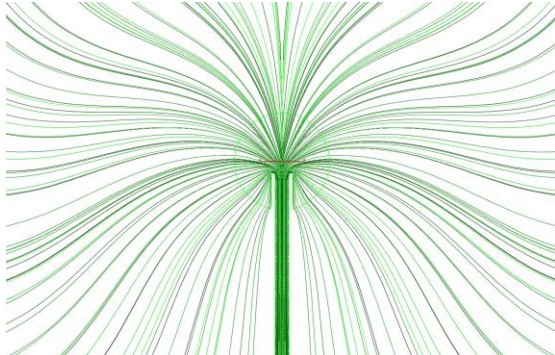

**Figure 9.** Cold flow streamlines from the porous burner for the conditions of normal flame No. 1 (19115B1).

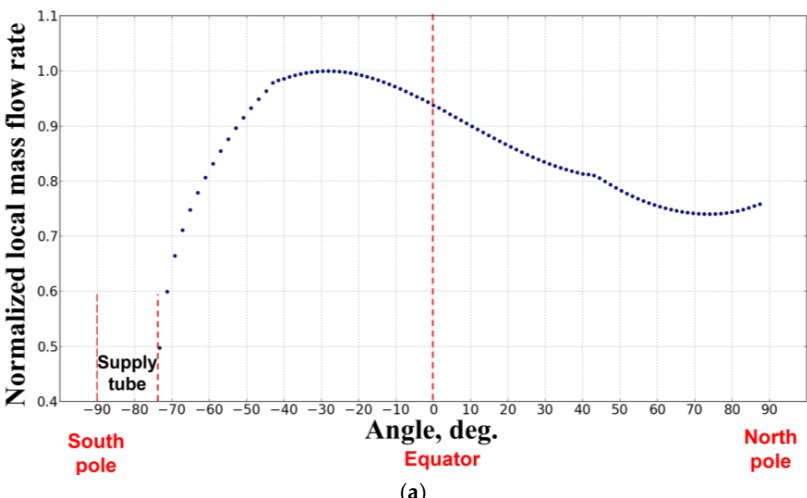

(a)

**Figure 10.** *Cont.*

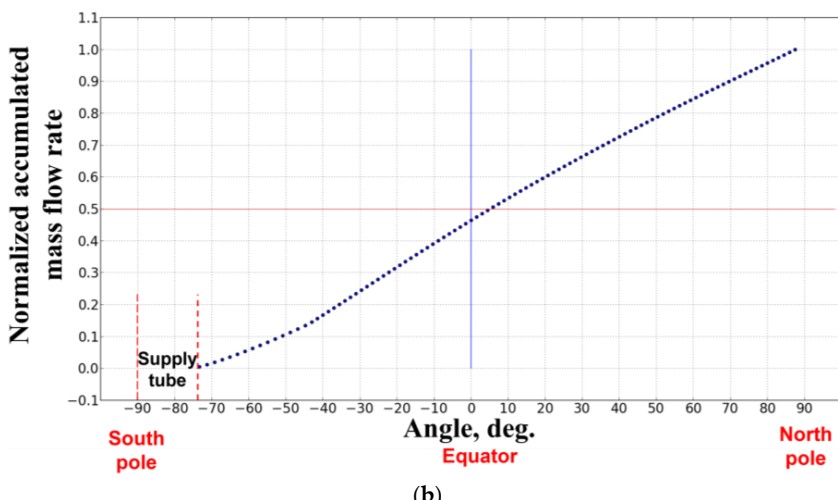

(**b**)

**Figure 10.** The calculated cold-flow dependences of the normalized local mass flow rate (**a**) and normalized accumulated mass flow rate of fuel (**b**) through the porous burner on the angle in the polar section of the sphere: $-90°$ corresponds to the south pole, $0°$ corresponds to the equator, and $+90°$ corresponds to the north pole of the sphere (normal flame No. 1 (19115B1)).

Reactive Flow

Figures 11 and 12 show the calculated evolution of the 2D normal flame No. 1 (19115B1) with the isothermal (Figure 11) and adiabatic (Figure 12) supply tubes. Each frame shows the value of time after ignition and the maximum gas temperature corresponding to the red color scale. In general, similar to 1D calculations, the flame radius increases and the flame temperature decreases with time. However, contrary to our expectations based on the experiments and 1D calculations, these changes are oscillatory rather than gradual and are accompanied by slow alterations in flame shape. The characteristic time period of oscillations is about 7 s. The flame in Figure 11 with the isothermal supply tube looks more corresponding to the experimental observations, as it exhibits the characteristic asymmetry in both flame shape and temperature distribution (luminosity) observed in the space experiment. It could be thus assumed that flame quenching near the gas supply tube observed in the experiments is caused by the cold gas supply tube. Flame shape alterations are caused by the incepience of two torroidal vortices of strength variable with time, as illustrated by Figure 13. The possible reason for vortex generation and slow flame oscillations is the shear flow due to the asymmetry in the mass flow rate of fluid supplied through the PB. Note that flame oscillations were not suppressed, neither by changes in the computational grid and differencing schemes nor by changes in the ignition procedure to make it as soft as possible. Interestingly, the oscillations were completely suppressed when we applied microgravity to the flow field, with the microgravity vector directed downstream of the supply tube. For the sake of illustration, Figure 14 shows the calculated evolution of the 2D inverse flame No. 25 (21328D1) with an isothermal supply tube and a microgravity of $0.01g$. As is seen, the flame shape is completely established 3 s after ignition. Due to the small but nonzero microgravity, hot combustion products are seen to move in the direction opposite to the gravity vector and stabilize the flame shape. Interestingly, in this case, the mean flame radius remains close to that obtained in the 1D calculation.

Video records of some inverse flames show very similar features. As an example, Figure 15 shows the evolution of the highly sooty inverse flame with an experiment duration of about 7 s. The existence of a soot cloud helps in visualizing the arising convective flow. Arrows in the figure show the positions of two concavities on the soot-cloud edge. Clearly, there is a definite directional motion of these concavities from left to right, i.e., upstream of the PB supply tube. This motion could be caused by the elevated mass flow rate of the oxidizer through the southern hemisphere of the PB (see the previous section), by the microgravity vector directed downstream of the supply tube, or by the combination of both effects.

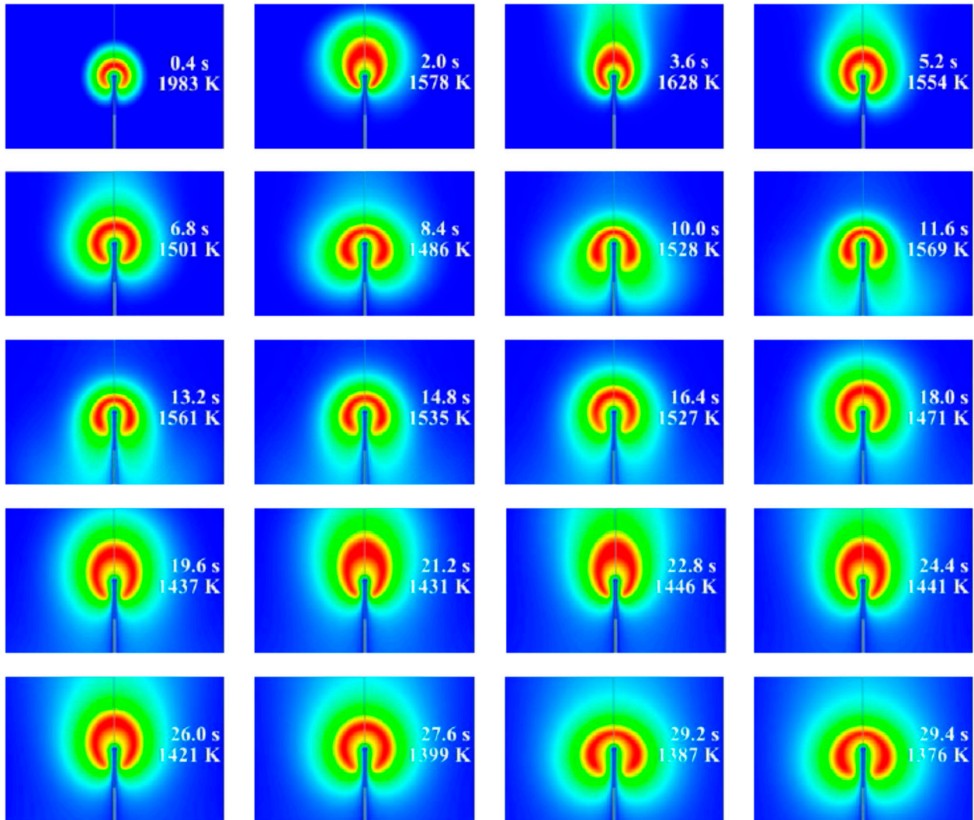

**Figure 11.** The calculated evolution of normal flame No. 1 (19115B1) with the isothermal supply tube.

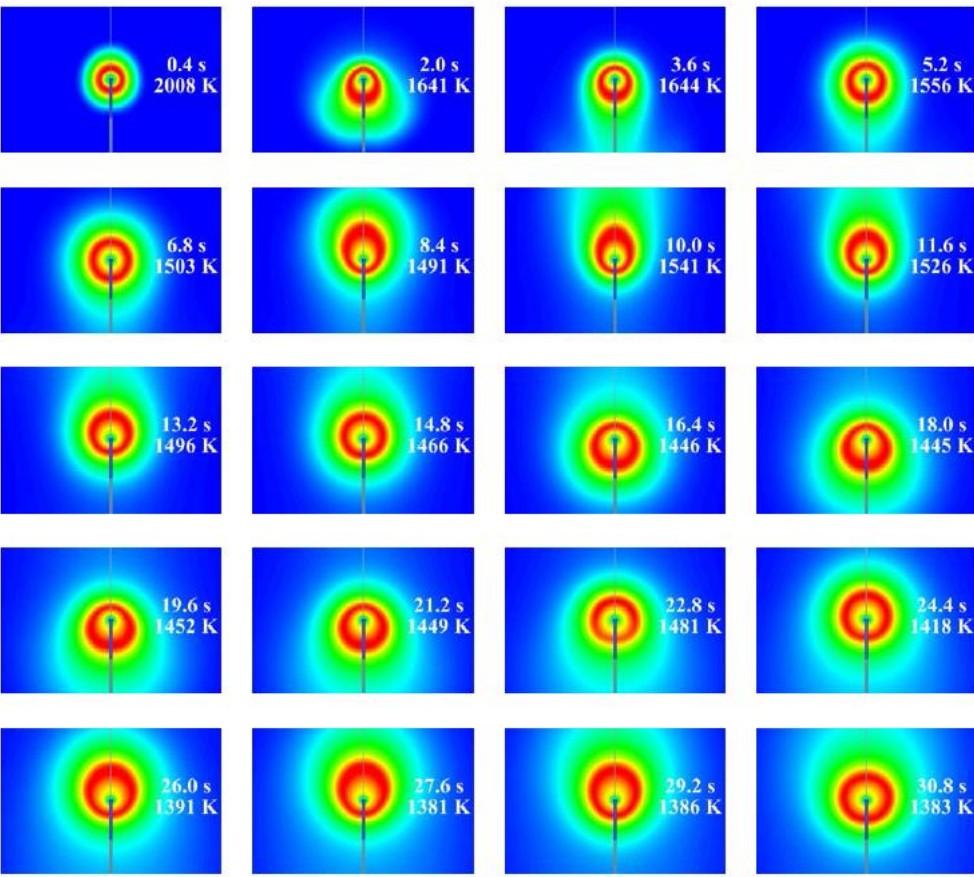

**Figure 12.** The calculated evolution of normal flame No. 1 (19115B1) with the adiabatic supply tube.

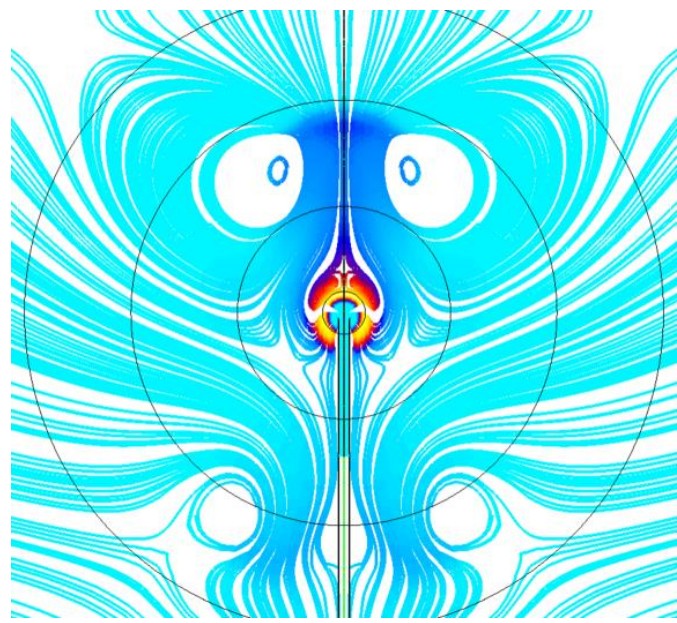

**Figure 13.** Flow streamlines with two torroidal vortices for the normal SDF.

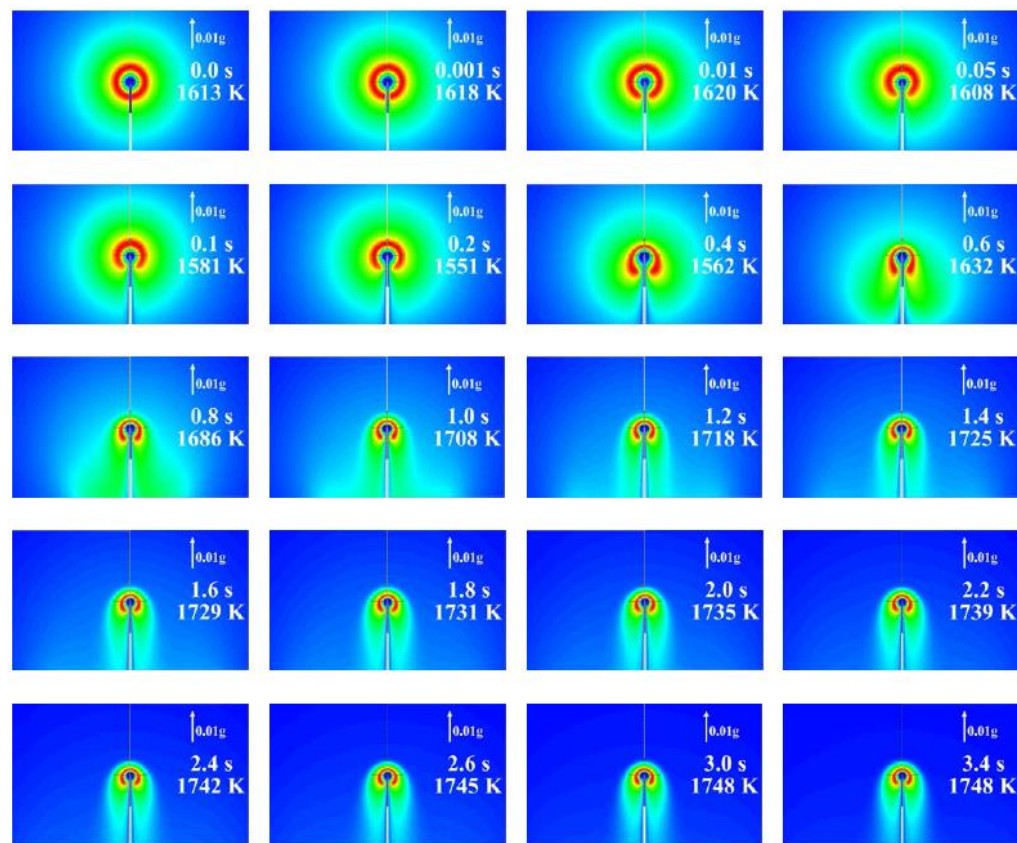

**Figure 14.** The calculated evolution of inverse flame No. 25 (21328D1) with an isothermal supply tube and a microgravity of 0.01*g*.

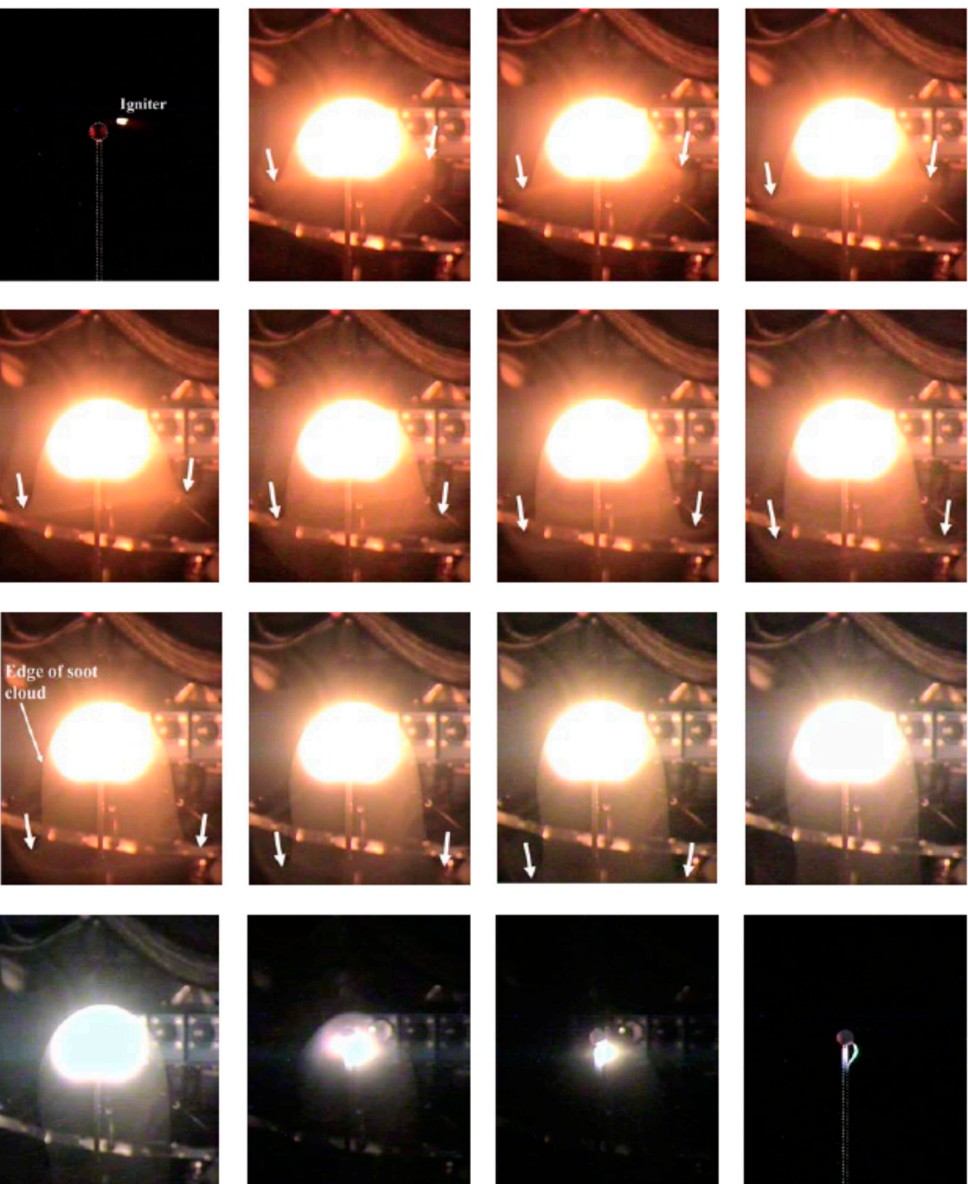

**Figure 15.** Video frames of the evolution of a highly sooty inverse flame. Arrows show the motion of some characteristic points at the soot cloud edge.

## 4. Conclusions

The evolution of 24 normal and 13 inverse spherical diffusion flames of ethylene under microgravity conditions was analyzed using 1D and 2D numerical simulations. The calculations were based on a 3D nonstationary model of the diffusion combustion of gases with the detailed kinetics of ethylene oxidation, supplemented by a macrokinetic mechanism of soot formation. The objective was to find some generalities in the evolution of normal and inverse flames in microgravity conditions and to reveal the possible reasons for the flame shape and luminosity asymmetry observed in the space experiment at the ISS. The following new results were obtained:

(1) There exist the unambiguous dependences of the ratio of flame radius to fluid mass flow rate through the PB, $R_f/G_{in}$, on the stoichiometric mixture fraction $Z_{st}$ for normal and inverse flames;

(2) The growth rate of normal flames decreases with $Z_{st}$ and they become more prone to be stationary. At $Z_{st} = 0.36$–$0.411$, the normal flames become almost stationary, i.e., their growth rates become very low;

(3) Contrary to normal flames, the growth rate of inverse flames increases with $Z_{st}$ and they become less prone to be stationary. At $Z_{st} = 0.203$–$0.218$, the inverse flames become almost stationary, i.e., their growth rates become very low;

(4) The maximum growth rate of inverse flames is considerably less than that of normal flames. This means that the lifetime of inverse flames must be longer than that of the normal flames at the identical value of $R_f/G_{in}$;

(5) At the same extinction radius, the extinction of the inverse flame occurs later than the extinction of the normal flame due to the lower growth rate;

(6) The flame radius at extinction is approximately constant in the wide range of $G_{in}$. A simple scaling law from the literature underestimates the flame radius at extinction;

(7) The 2D cold-flow calculations show that the supply tube blocks a part of the surface of the porous burner in the vicinity of the south pole. Due to this blockage, the normalized accumulated mass flow rate of fluid through the southern hemisphere is about 15% less than through the northern hemisphere. The local mass flow rate of fluid through the porous burner is nonuniform with the maximum flow rate attained in the angular interval $[-20°, -40°]$ in the southern hemisphere;

(8) The cold isothermal gas supply tube looks more corresponding to the experimental observations than the adiabatic supply tube, as it exhibits the characteristic asymmetry in both flame shape and temperature distribution (luminosity) observed in the experiments. It could be thus assumed that flame quenching near the gas supply tube observed in the experiments is caused by the cold gas supply tube;

(9) The 2D calculations reveal the oscillatory evolution of ethylene diffusion flames with slow alterations in flame shape and temperature caused by the incepience of torroidal vortices in the surrounding gas;

(10) The introduction of directional microgravity on the level of $0.01g$ allows complete suppression of flame oscillations.

The obtained results are important for both better understanding the mechanisms of diffusion combustion in microgravity conditions and improving practical means to avoid and fight possible fire accidents on spacecraft. As for the latter, they are usually based on the knowledge obtained in laboratories in the conditions of terrestrial gravity and on imaginary scenarios for onboard fire development. Future work will be focused on the 3D simulation of normal and inverse spherical diffusion flames with the localized (rather than spherically symmetrical) ignition and complex thermal boundary conditions on the gas supply tube (conjugate heat transfer rather than an adiabatic or isothermal wall).

**Author Contributions:** Conceptualization, S.M.F.; methodology, S.M.F. and V.S.I.; investigation, S.M.F., V.S.I., F.S.F. and I.V.S.; data curation, V.S.I. and F.S.F.; writing—original draft preparation, S.M.F.; writing—review and editing, S.M.F. All authors have read and agreed to the published version of the manuscript.

**Funding:** This research was partly funded by the Russian Space Agency (Adamant project) and by a subsidy given to the Federal State Institution "Scientific Research Institute for System Analysis of the Russian Academy of Sciences" to implement the state assignment on topic No. FNEF-2022-0005 (Registration No. 1021060708369-1-1.2.1).

**Institutional Review Board Statement:** Not applicable.

**Informed Consent Statement:** Not applicable.

**Data Availability Statement:** Research data can be provided upon request.

**Acknowledgments:** Fruitful collaboration with US colleagues R. Axelbaum, P.B. Sunderland, P.H. Irace; G. Yablonsky, K. Waddell, and K. Minhyeng within the joint Flame Design–Adamant project is gratefully acknowledged.

**Conflicts of Interest:** The authors declare no conflict of interest.

## Nomenclature

| | |
|---|---|
| $a_l$ | Emissivity of the $l$th emitting gas |
| $A_k$ | Pre-exponential factor, ($k = 1, \ldots, L$) |
| $c_{p,l}$ | Specific heat at constant pressure, ($l = 1, \ldots, N$) |
| $c_s$ | Solid skeleton heat capacity |
| $(C/O)_c$ | Threshold local C/O atomic ratio |
| $d$ | Characteristic size of the solid skeleton |
| $d_{soot}$ | Conditional soot particle size |
| $D_l$ | Effective diffusion coefficient of the $l$th species, ($l = 1, \ldots, N$) |
| $E_k$ | Activation energy, ($k = 1, \ldots, L$) |
| $\left(\frac{\partial p}{\partial x_i}\right)_s$ | added momentum source in a porous medium |
| $G_{in}$ | Inlet mass flow rate |
| $g_i$ | $i$th component of the vector of acceleration of gravity $g$; |
| $H$ | Mean gas static enthalpy |
| $H_l^0$ | Standard enthalpy of formation of the $l$th species, ($l = 1, \ldots, N$) |
| $I$ | Mean gas total enthalpy |
| $j_l$ | Molecular mass flux of the $l$th species, ($l = 1, \ldots, N$) |
| $j_{lj}^t$ | Turbulent mass flux of the $l$th species, ($l = 1, \ldots, N$), ($j = 1, 2, 3$) |
| $L$ | Total number of chemical reactions in the gas |
| $n_k$ | Temperature exponent, ($k = 1, \ldots, L$) |
| $N$ | Number of gas species |
| $P$ | Mean gas pressure |
| $P_0$ | Initial pressure |
| $q_j$ | Molecular heat flux, ($j = 1, 2, 3$) |
| $q_j^t$ | Turbulent heat flux, ($j = 1, 2, 3$) |
| $Q$ | Mean source of energy due to chemical transformations |
| $R_0$ | Length of the buffer channel |
| $R_f$ | Flame radius |
| $R_s$ | Radius of porous sphere |
| $r_\infty$ | Radius of the outer wall of the chamber |
| $R$ | Universal gas constant |
| $S_{in}$ | Passage area of gas supply tube |
| $S_{PB}$ | Specific surface area of the porous burner |
| $S_{soot}$ | Specific emitting surface area |
| $t$ | Time |
| $t_{ign}$ | Time of ignition |
| $T$ | Temperature |
| $T_0$ | Initial temperature |
| $T^0$ | Standard temperature |
| $T_c$ | Threshold local temperature of soot formation |
| $T_{ign}$ | Ignition temperature |
| $T_s$ | Temperature of porous sphere |
| $u_i$ | Superficial velocity inside the porous medium, ($j = 1, 2, 3$) |
| $U_i$ | The $i$th component of the mean gas velocity vector, ($j = 1, 2, 3$) |
| $V$ | Chamber volume |
| $\dot{w}_l$ | Mean source of mass due to chemical transformations, ($l = 1, \ldots, N$) |
| $W$ | Molecular mass |
| $W_O$ | Molecular mass of oxidizer |
| $W_F$ | Molecular mass of fuel |
| $x_j$ | Cartesian coordinate, ($j = 1, 2, 3$) |
| $X_l$ | Volume fraction of the $l$th emitting gas |
| $Y_{i0}$ | Initial species mass fractions, ($i = 1, \ldots, N$) |
| $Y_{i,in}$ | Inlet species mass fractions, ($i = 1, \ldots, N$) |
| $Y_l$ | Mean mass fraction of the $l$th species, ($l = 1, \ldots, N$) |

| | |
|---|---|
| $Y_{soot}$ | Soot mass fraction |
| $Z_{st}$ | Stoichiometric mixture fraction |
| $\alpha_s$ | Heat transfer coefficient between gas and porous sphere |
| $\delta_s$ | Delta function |
| $\varepsilon_s$ | Coefficient of radiation absorption by the porous sphere material |
| $\kappa$ | Permeability |
| $\lambda$ | Thermal conductivity of the $l$th species |
| $\lambda_s$ | Solid skeleton thermal conductivity |
| $\mu$ | Dynamic viscosity of gas |
| $\nu_O$ | Stoichiometric coefficient of oxidizer in the overall reaction equation |
| $\nu_F$ | Stoichiometric coefficient of fuel in the overall reaction equation |
| $v'_{l,k}$ | Stoichiometric coefficients of the $l$th species in the reactants of the $k$th reaction, $(l = 1, \ldots, N)$, $(k = 1, \ldots, L)$ |
| $v''_{l,k}$ | Stoichiometric coefficients of the $l$th species in the products of the $k$th reaction, $(l = 1, \ldots, N)$, $(k = 1, \ldots, L)$ |
| $\rho$ | Mean gas density |
| $\rho_s$ | Solid skeleton density |
| $\rho_{soot}$ | Soot density |
| $\sigma$ | Stefan–Boltzmann constant |
| $\tau_{ij}$ | Tensor of viscous stresses |
| $\tau^t_{ij}$ | Tensor of turbulent stresses |
| $\varphi$ | Porosity |
| $\Psi_s$ | Added heat source in porous medium |
| $\Omega$ | Heat source/sink other than that of chemical nature |
| $\Omega_s$ | Heat source/sink for porous sphere |
| $\Omega_{sg}$ | Radiation absorption |

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
