# Peer review of "Spherical Diffusion Flames of Ethylene in Microgravity: Multidimensional Effects"

_fire, doi:10.3390/fire6080285_

Round 1

Reviewer 1 Report

The reviewer thanks the authors for submission of the manuscript. The manuscript describes a numerical investigation of a set of diffusion flame experiments performed in microgravity. The thesis of the manuscript is clearly stated and appropriately cited. The geometry described of a diffusion flame around a spherical porous burner is an important fundamental flame geometry that can be used for the study of extinction mechanisms. The fidelity of the numerical simulations was both low (1D) and high (2D) to provide insight into the lack of true spherical symmetry found in the experiments.

I recommend expanding the discussion of results and the analysis on the 1D flame cases. The objective of the 1D simulations is stated on manuscript lines 332-333 to examine and generalize normal/inverse flames, but the results/discussion is largely limited to the observable (flame radius / boundary velocity) vs (stoichiometric mixture fraction) at different points in time. The discussion of comparing the extinction radius to an analytical theory is limited to a single paragraph and figure (lines 407-416).

For low dimensional simulations there are many parametric studies (e.g. using nondimensional parameters) that can be taken to fundamentally examine the family character of normal/inverse geometries, e.g. : (1) simplified 1 step chemistry to examine the importance of chemistry (Damkoehler number), (2) changing the coordinate system to cartesian to examine the importance of the flame front curvature (Markstein number), (3) changing the thermal-diffusive properties of the mixture (Lewis number).

I recommend expanding the discussion and comparison of the 2D results to the 1D results. The discussion of the 2D flame cases is limited to a single paragraph (lines 440-470) with no figures shown to support 1D/2D comparisons stated in the text. The non-symmetries in system are discussed, but the significance of the non-ideality is not placed in context of the thesis (extinction limits and normal/inverse character) using results from the 1D simulations. 

Additional comments:

lines 111-114, the experimental definition used for flame radius is the average of the luminous zone (chemiluminescence + soot emission) while the numerical definition is the location of max temperature. The traditional definition for numerical flame position with respect to chemiluminescence is more closely tied to the location of peak heat release, though post processing data sets adding the kinetics for OH*/CH*/C2* emission is more defensible (e.g. mechanisms in https://doi.org/10.1016/S0010-2180(02)00399-1). Similar post-processing can be done to prediction the location of peak soot radiation.

lines 223-224, please elaborate why N2 and O2 treated as radiating species. Readers are most likely familiar with flame radiation models (manuscript source 32) that do not consider these species.

lines 318-322, manuscript states that in figure 4a and 5a the predicted flame extinction occurs as a kink in the flame radius vs time curve, and that the kink lines up with the last experimental data point for each figure. This is true for figure 4a at 41 sec, but appears to be false in figure 5a for 17 sec experimental vs 27 sec predicted.

Author Response

We are grateful to the reviewer for valuable comments. We have made our best to follow all the comments. All changes in the revised manuscript are marked in yellow.

The reviewer thanks the authors for submission of the manuscript. The manuscript describes a numerical investigation of a set of diffusion flame experiments performed in microgravity. The thesis of the manuscript is clearly stated and appropriately cited. The geometry described of a diffusion flame around a spherical porous burner is an important fundamental flame geometry that can be used for the study of extinction mechanisms. The fidelity of the numerical simulations was both low (1D) and high (2D) to provide insight into the lack of true spherical symmetry found in the experiments.

I recommend expanding the discussion of results and the analysis on the 1D flame cases. The objective of the 1D simulations is stated on manuscript lines 332-333 to examine and generalize normal/inverse flames, but the results/discussion is largely limited to the observable (flame radius / boundary velocity) vs (stoichiometric mixture fraction) at different points in time. The discussion of comparing the extinction radius to an analytical theory is limited to a single paragraph and figure (lines 407-416). For low dimensional simulations there are many parametric studies (e.g. using nondimensional parameters) that can be taken to fundamentally examine the family character of normal/inverse geometries, e.g. : (1) simplified 1 step chemistry to examine the importance of chemistry (Damkoehler number), (2) changing the coordinate system to cartesian to examine the importance of the flame front curvature (Markstein number), (3) changing the thermal-diffusive properties of the mixture (Lewis number).

In our study, we used the detailed chemistry and transport models, which did not deal with characteristic Damkoehler, Markstein, and Lewis numbers. These numbers are difficult to determine as the role of different species and reactions as well as the flow pattern changes with time and location for the flames under consideration. Using our approach, we succeeded in extracting some new general features for many long-lived (up to 200 s) normal and inverse flames in microgravity, which we discuss in Section 3.2.1. Note that the results of most existing simplified models are compared with the results of short-time (2.2 s) experiments in drop-towers. Among other things, we have found in [21] that the results of such short-time experiments are ignition dependent, that is the yields of soot and other products depend on the ignition stage and cannot be considered as solely flame borne. The ignition stage appeared to be “memorized” by the flame for about 10 s! Nevertheless, some simplified models available in the literature provided a reasonable qualitative and even quantitative agreement with these short-time experiments. In view of it, to address this comment we have added the following sentences to the Introduction section:

“Most of the experimental studies were carried out in drop-towers with a short observation time, insufficient for the flame to develop independently of the ignition stage. Most of the theoretical and computational studies used simplified descriptions of transport and chemistry with some characteristic Damkoehler, Markstein, and Lewis numbers that could yield qualitatively loose and quantitatively inaccurate results [10].”

I recommend expanding the discussion and comparison of the 2D results to the 1D results. The discussion of the 2D flame cases is limited to a single paragraph (lines 440-470) with no figures shown to support 1D/2D comparisons stated in the text. The non-symmetries in system are discussed, but the significance of the non-ideality is not placed in context of the thesis (extinction limits and normal/inverse character) using results from the 1D simulations. 

We use the same code and the same settings in both 2D and 1D calculations. If we do not include the gas supply tube into consideration and solve the spherically symmetrical problem, the 2D and 1D results are IDENTICAL. The only difference between 2D and 1D calculations considered in Section 3.2.2 is the extension of the 2D computational mesh to include the gas supply tube, while the mesh around the porous burner is essentially the same. Nevertheless, we showed that 2D flames with the gas supply tube attached to the porous burner exhibited slow oscillations at zero gravity, which was not the case with 1D calculations. Therefore, there was no sense to directly compare 2D (oscillatory) and 1D (nonoscillatory) calculations. We attributed these oscillations of 2D flames to the vortical structures formed around the porous burner due to the asymmetry in the mass flow rate of the injected fluid in the southern hemisphere. Adoption of small (nonzero) gravity suppressed these oscillations, while the mean flame radius remained close to that obtained in the 1D calculation.

To avoid misunderstanding, we have added several new sentences at the beginning and at the end of Section 3.2.2:

“As mentioned earlier in this paper, for 2D calculations we use the same code and the same settings as in 1D calculations. Therefore, the results of 2D and 1D calculations for the spherically symmetrical statement of the problem are identical. The only difference between 2D and 1D calculations discussed below in this paper is the extension of the 2D computational mesh to include the gas supply tube, while the mesh around the PB is essentially the same.”

“The possible reason for vortex generation and slow flame oscillations is the shear flow due to the asymmtry in the mass fow rate of fluid supplied through the PB. Note that flame oscillations were not suppressed neither by changes in the computational grid and differencing schemes nor by changes in the ignition procedure to make it as soft as possible.”

“Interestingly, in this case the mean flame radius remains close to that obtained in the 1D calculation.”

Additional comments:

lines 111-114, the experimental definition used for flame radius is the average of the luminous zone (chemiluminescence + soot emission) while the numerical definition is the location of max temperature. The traditional definition for numerical flame position with respect to chemiluminescence is more closely tied to the location of peak heat release, though post processing data sets adding the kinetics for OH*/CH*/C2* emission is more defensible (e.g. mechanisms in https://doi.org/10.1016/S0010-2180(02)00399-1). Similar post-processing can be done to prediction the location of peak soot radiation.

In our previous article [21] published in Mathematics (https://doi.org/10.3390/math11020261), we have used the same definition of flame radius. In [21], we have presented the detailed structures of typical normal and inverse flames in terms of the spatial distributions of temperature and species concentrations (see Fig. 7 in the cited article) indicating that the temperature maximum corresponds well with the maxima of CO, CO2, H2O and OH concentrations. As for soot, its maximum concentration is attained either inside the flame (for normal flames) or outside the flame (for inverse flames). The reaction mechanism used in our studies does not include electronically excited species like OH*, CH*, and C2*. Therefore, we cannot identify the flame radius by their maximum concentrations. Nevertheless, if one assumes that the local concentration of excited OH* is proportional to the local concentration of OH in the base state than it appears that the local maximum of OH concentration corresponds well with the maximum temperature in the flame. To address this comment, we have repeatedly included the reference to [21] in the sentence dealing with the definition of flame radius.

lines 223-224, please elaborate why N2 and O2 treated as radiating species. Readers are most likely familiar with flame radiation models (manuscript source 32) that do not consider these species.

We have included N2 and O2 as radiating species for the sake of generality, however their contribution to the radiative heat transfer appeared to be negligeable despite large concentrations. To address this comment, we have slightly modified the formulation of assumption #5.

lines 318-322, manuscript states that in figure 4a and 5a the predicted flame extinction occurs as a kink in the flame radius vs time curve, and that the kink lines up with the last experimental data point for each figure. This is true for figure 4a at 41 sec, but appears to be false in figure 5a for 17 sec experimental vs 27 sec predicted.

In the last sentence of Section 3.1 we only state that the radiative extinction in the experiment corresponds to the last experimental point, whereas the radiative extinction in the calculation corresponds to the kink at the curve. Several lines above this sentence we claim that “the 1D model of Section 2.1 predicts the FLAME RADII (not FLAME EXTINCTION TIME) at extinction reasonably well for both normal and inverse flames.” This means that when comparing the ordinates of the last experimental point and the kink on the curves in Figs. 4a and 5a, one observe a reasonable agreement. As a matter of fact, one cannot compare the measured and calculated extinction times because the ignition procedure is not modeled properly. To address this comment and to avoid misunderstanding, we have added a new sentence at the end of Section 3.1:

“The ordinates of the last experimental point and the kink at the curve in Figures 4a and 5a correspond reasonably well with each other.”

Reviewer 2 Report

This is a very interesting and well-written paper that surely deserves publication in Fire. I have only the following comments that the authors should implement in the revised manuscript prior to publication.

Abstract - The abstract is too long and should be condensed.

Conclusions - The authors should better highlight the practical impact of the results obtained in this paper. They should also give a structured outlook on future research work.

Moderate editing of English language is required.

Author Response

We are grateful to the reviewer for valuable comments. We have made our best to follow all the comments. All changes in the revised manuscript are marked in green.

This is a very interesting and well-written paper that surely deserves publication in Fire. I have only the following comments that the authors should implement in the revised manuscript prior to publication.

Abstract - The abstract is too long and should be condensed.

We have shortened the Abstract by removing some general sentences and moving some sentences to the Introduction section.

Conclusions - The authors should better highlight the practical impact of the results obtained in this paper. They should also give a structured outlook on future research work.

We have added one paragraph at the end of Conclusions to highlight the practical impact of the obtained results and to provide the outlook on the future research work:

“The obtained results are important for both better understanding the mechanisms of diffusion combustion in microgravity conditions and improving practical means to avoid and fight possible fire accidents on spacecraft. As for the latter, they are usually based on the knowledge obtained in laboratories in the conditions of the terrestrial gravity, and on imaginary scenarios for onboard fire development. Future work will be focused on the 3D simulation of normal and inverse spherical diffusion flames with the localized (rather than spherically symmetrical) ignition and complex thermal boundary conditions on the gas supply tube (conjugate heat transfer rather than adiabatic or isothermal wall).”
